# How the Microbiota May Affect Celiac Disease and What We Can Do

**DOI:** 10.3390/nu16121882

**Published:** 2024-06-14

**Authors:** Mariarosaria Matera, Stefano Guandalini

**Affiliations:** 1Pediatric Clinical Microbiomics Service, Misericordia Hospital, Via Senese 161, 58100 Grosseto, Italy; mariarosaria.matera@uslsudest.toscana.it; 2Section of Pediatric Gastroenterology, Hepatology and Nutrition, Celiac Disease Center, University of Chicago Medicine, 5841 S. Maryland Ave. MC 4065, Chicago, IL 60637, USA

**Keywords:** celiac disease, gluten, microbiota, enterocytes, probiotics, dysbiosis

## Abstract

Celiac disease (CeD) is an autoimmune disease with a strong association with human leukocyte antigen (HLA), characterized by the production of specific autoantibodies and immune-mediated enterocyte killing. CeD is a unique autoimmune condition, as it is the only one in which the environmental trigger is known: gluten, a storage protein present in wheat, barley, and rye. How and when the loss of tolerance of the intestinal mucosa to gluten occurs is still unknown. This event, through the activation of adaptive immune responses, enhances epithelial cell death, increases the permeability of the epithelial barrier, and induces secretion of pro-inflammatory cytokines, resulting in the transition from genetic predisposition to the actual onset of the disease. While the role of gastrointestinal infections as a possible trigger has been considered on the basis of a possible mechanism of antigen mimicry, a more likely alternative mechanism appears to involve a complex disruption of the gastrointestinal microbiota ecosystem triggered by infections, rather than the specific effect of a single pathogen on intestinal mucosal homeostasis. Several lines of evidence show the existence of intestinal dysbiosis that precedes the onset of CeD in genetically at-risk subjects, characterized by the loss of protective bacterial elements that both epigenetically and functionally can influence the response of the intestinal epithelium leading to the loss of gluten tolerance. We have conducted a literature review in order to summarize the current knowledge about the complex and in part still unraveled dysbiosis that precedes and accompanies CeD and present some exciting new data on how this dysbiosis might be prevented and/or counteracted. The literature search was conducted on PubMed.gov in the time frame 2010 to March 2024 utilizing the terms “celiac disease and microbiota”, “celiac disease and microbiome”, and “celiac disease and probiotics” and restricting the search to the following article types: Clinical Trials, Meta-Analysis, Review, and Systematic Review. A total of 364 papers were identified and reviewed. The main conclusions of this review can be outlined as follows: (1) quantitative and qualitative changes in gut microbiota have been clearly documented in CeD patients; (2) intestinal microbiota’s extensive and variable interactions with enterocytes, viral and bacterial pathogens and even gluten combine to impact the inflammatory immune response to gluten and the loss of gluten tolerance, ultimately affecting the pathogenesis, progression, and clinical expression of CeD; (3) gluten-free diet fails to restore the eubiosis of the digestive tract in CeD patients, and also negatively affects microbial homeostasis; (4) new tools allowing targeted microbiota therapy, such as the use of probiotics (a good example being precision probiotics like the novel strain of *B. vulgatus* (20220303-A2) begin to show exciting potential applications.

## 1. Introduction

The most common genetically induced adverse reaction to food, Celiac disease (CeD) is an autoimmune disease strongly associated with specific HLA (human leukocyte antigen) class II haplotypes and characterized by the production of specific autoantibodies. CeD is a unique autoimmune condition, as it is the only one in which the environmental trigger is known: gluten, a storage protein present in wheat, barley, and rye.

The disease affects approximately 1% of the world population and its incidence has been progressively increasing annually by about 8% with a female predominance (1.5 times more frequent). The global prevalence of celiac disease is 1.4% when assessed by serological markers, and 0.7% as biopsy-confirmed. There are also important geographical variations in prevalence, being highest in Europe and Oceania (0.8%) and lowest in South America (0.4%) [1].

The loss of tolerance to gluten occurs only in genetically predisposed individuals and may occur at any time after birth. The HLA class II molecules DQ2 and DQ8, expressed on the surface of antigen-presenting cells, are the main genetic risk factors for CeD, but genome-wide association studies (GWAS) have identified as many as 39 non-HLA risk loci, among which we recognize genes involved in immune function which include the genes for interleukin 18 and 21 (IL18 and IL21), for chemokine receptors (CCR1, CCR2, CCR5), and genes associated with alteration of intestinal permeability (MYO9B, PARD3 and MAG12). Furthermore, in 2016 new genes have been identified [2] involved in the pathogenesis of CeD (TAGAP, IL18R1, RGS21, PLEK, and CCR9) and in particular the regions CTLA4 and LPP were found to be associated with anti-tissue transglutaminases. Of note, these genes also appear to be correlated with the type of bacterial colonization and the composition of the intestinal microbiota [3,4,5]. However, genetic predisposition alone is not sufficient to bring about the onset of the disease, as it requires the involvement of environmental factors.

Obviously, the only necessary single one is gluten, a heterogeneous mixture of proteins present in wheat, barley, and rye. Gliadin and glutenin, the main components of gluten, are responsible for the characteristic viscous-elastic consistency and cohesiveness imparted to doughs and making them perfectly apt for leavening and bread-making. 

In fact, multiple additional environmental factors also act on epigenetic programming through changes in the composition of the intestinal microbial ecosystem. This is indeed known to actively participate in the maturation of epithelial barriers and in the gut-associated lymphoid tissue (GALT), as well as in the homeostasis of innate and adaptive immunity and in initiation of tissue repair programs, all factors implicated in impaired tolerance to dietary gliadin [6,7].

As is well known, treatment of CeD relies exclusively, so far, on a strict gluten-free diet (GFD), that in the vast majority of patients is capable of reversing the immune-mediated intestinal damage and of restoring intestinal digestive-absorption functions.

The GFD, however, needs to be nutritionally balanced and supplemented with other cereals, micronutrients, and dietary fiber in order to avoid the nutritional deficiencies often described in patients on GFD, in part as a result of an excessive use of ultra-processed products [8,9].

In this review covering relevant papers from 2010 to 2024, we examined recent progresses with respect to the relationship between gluten and the oro-gastro-intestinal microbiota, the effect of GFD on the microbiota, and finally explore the potential mechanistic use of precision microecological agents in the microbiota-targeted prevention and treatment of CeD.

## 2. Gluten and the Pathophysiology of Celiac Disease

The main antigenic proteins of gluten consist of monomeric subunits (gliadins), grouped into four fractions based on the molecular weight and electrophoretic response (α, β, γ, and ω—of which, α, γ, and ω are associated with CeD) and polymeric subunits (glutenins) with a high content of polypeptide residues rich in glutamine and proline. Most gluten is hydrolyzed by pepsin present in the stomach into high molecular weight immunogenic polymeric peptides rich in proline. Such peptides are not susceptible to proteolysis by human gastric and pancreatic enzymes nor by brush-border bound peptidases. Therefore, these polymeric peptides tend to accumulate in the small intestinal lumen where they can remain for a long time. Well-known examples of highly immunogenic and protease-resistant peptides are the 33-mer peptide derived from α-gliadins and the 26-mer peptide derived from γ-gliadins. Additionally, the P57-P89 peptide and the P31-P43 peptide are able to trigger the innate immune response, with a primary role played by IL-2 mediating an increase in intraepithelial lymphocytes, as well as an adaptive immune response mediated by CD4+Th1 cells [8,10,11,12].

Furthermore, we have come to learn that various intestinal bacteria, both commensals and opportunistic pathogens, produce a wide range of proteases that can degrade dietary components and host proteins, impacting immune homeostasis and the inflammatory and functional state of the host as we will further explore later [1,8,13].

A fraction of the ingested gluten peptides can then translocate from the intestinal lumen, through the intestinal epithelial barrier, via mostly paracellular but also intracellular pathways until they reach the lamina propria. The intestinal epithelium in physiological conditions is mostly impermeable to macromolecules, but in celiac disease the integrity of the intestinal epithelial barrier is compromised due to alteration of the tight junction (TJ), thereby allowing gluten peptides to more easily reach the lamina propria. Here, tissue transglutaminase 2 (tTG2) deamidates them, thus creating negatively charged, highly immunogenic deamidated gliadin peptides (DGPs). These show a high binding affinity with HLA-DQ2/DQ8 molecules expressed on dendritic cells; HLA-gliadin peptide complexes are thus formed and then presented to naïve CD4+ T cells, thus enhancing a gluten-specific T helper 1 (Th1) inflammatory response, characterized by the production of high levels of interferon gamma (IFN-γ) and interleukin (IL)-21 in the small intestinal lamina propria. The gluten-specific CD4+ T cell response is maintained by several cytokines, and especially IL-15 and IL-21, that synergistically promote gluten-mediated increase of IFN-γ production. 

Interestingly, the triggering of a gluten-specific CD4+ T cell response might be brought about by some viral infections, as exemplified by Reoviruses, but also possibly by bacterial antigens. 

In summary, the pathogenetic changes observed in celiac disease include release of pro-inflammatory cytokines, activation of IEL and the production of specific antibodies (schematically illustrated by Figure 1). This succession of events results ultimately in inflammatory changes in the mucosa of the small intestine leading to enterocytes’ apoptosis, increased IEL (>25/100 enterocytes), crypt hyperplasia, and villous atrophy, all well-known landmarks of celiac disease [12,14,15,16]. What then, if any, is the role of microbiota in this series of events? 


**Pathogenesis of CeD**


Loss of tolerance to ingested gluten occurs only in individuals expressing HLA class II DQ2 and/or DQ8 molecules and exposed to multiple hits from environmental factors. Among them, premature birth, delivery by caesarean section, absence of breastfeeding, early exposure to infectious agents, and antibiotics can also act on epigenetic programming through changes in the composition of the gut microbial ecosystem that can interfere with the maturation of the intestinal barrier, gut-associated lymphoid tissue (GALT), and the homeostasis of innate and adaptive immunity.

Gluten is hydrolyzed by pepsin into immunogenic polymeric peptides that accumulate in the lumen of the small intestine. Not being further hydrolyzed by pancreatic or brush border-associated peptidases, they can then translocate through the permeable intestinal epithelial barrier into the lamina propria where tTG2 transforms them into highly immunogenic deamidated gliadin peptides (DGP). These show a high binding affinity with HLA-DQ2/DQ8 molecules expressed on dendritic cells. HLA-DGP peptide complexes are presented to naïve CD4+ T cells that repectively differentiate into the following: (1)CD8+ cells, which are cytotoxic cells becoming intraepithelial lymphocytes (IEL) that participate in enterocyte apoptosis.(2)Th17 inflammatory cells that, like Th1 cells, produce IL-21, IFN-γ, and TNF a.(3)Th1 inflammatory cells producing high levels of IFN-γ, IL-21, and TNF-a.TNF-a stimulates the production of IL-12 and IL-15 which synergistically promote the increase in gluten-mediated IFN-γ production thus further pushing infiltration of IEL promoting enterocyte apoptosis, crypt hyperplasia, and villous atrophy,(4)Th2 cells. Through the production of IL-4 and IL-6, they stimulate the progression of B-cells into plasma cells and the production of specific autoantibodies against endomysium, gliadin, and transglutaminase, thus participating in intestinal damage.


**Protective Factors and Targeted Microbiota Intervention**


(1)The use of a Mediterranean-type diet during the first 2 years of life has been shown to have a protective effect in preventing the development of CeD. Such a diet is rich in fiber and phytochemicals capable of stimulating the intestinal growth of eubiotic commensal bacteria that produce adequate proportions of short-chain fatty acids (SCFAs). SCFAs act by counteracting intestinal permeability (increasing mucus turnover and increasing the expression of tight junctions) and also by modulating the immune system. In fact, they promote homeostasis favoring Tregs that produce IL-10 and counteract both Th1 cells and the production of autoantibodies.(2)“Precision” probiotics are capable of several effects all blunting the inflammatory changes seen in CeD. In fact, they fight pathogenic and inflammatory species, restore eubiotic species producing SCFAs, produce peptidases capable of degrading immunogenic gliadin peptides, promote immune homeostasis through the enhancement of Tregs, modulate the permeability of the intestinal barrier, and produce Aryl receptor (AhR) ligands correlated with increased IL-22, intestinal stem cell proliferation, and restoration of intestinal mucosal damage.(3)Postbiotics—though so far less investigated in this regard—have the potential to improve gut barrier function by increasing tight junction expression and preventing the inflammatory effects of gliadin.

## 3. How the Gut Microbiota Is Made and Its Relationship with CeD

The human body is inhabited by bacteria, archaea, fungi, and viruses. Bacteria alone amount to 100 trillion with a ratio of 1.3:1 compared to our own cells. In humans, the gut site represents the living niche with the highest microbial concentration up to 10^12^ bacteria per gram of feces.

Bacteria are divided into nine main macrogroups defined as *phyla: Actinobacteria, Bacteroidetes*, *Cyanobacteria, Firmicutes*, *Fusobacteria*, *Lentisphaerae*, *Proteobacteria*, *Tenericutes*, and *Verrucomicrobia.*

*Firmicutes* and *Bacteroidetes* are by far the physiologically predominant *phyla* in the gut fecal microbiota of mature subjects as these two *phyla* alone constitute approximately 90% of the entire intestinal microbial structure. However, the composition of the microbiota differs, in every individual, based on genetics and lifestyle. Furthermore, in the various parts of the gastrointestinal tract the microbial composition is influenced by pH, gastrointestinal secretion, availability of energy substrate, and quantity of oxygen which determine a progressive cranio-caudal decrease in aerobic bacterial species and an increase in strictly anaerobic bacterial species. Consequently, the profile of the gut microbiota associated with health and/or disease, including CeD, depends on the specific niche or sample investigated.

The intestinal microbiota plays relevant roles in human physiology: it is involved in the differentiation of the intestinal epithelium, opposes pathogenic colonization, regulates intestinal permeability, triggers immunity to respond to antigenic stimulation, produces vitamins, hormones and neurotransmitters, contributes to digestion of food, modulates the entire energy metabolism of the host, and participates in the communication of the gut–brain axis, thus helping to maintain a state of health. It is therefore clear that the alteration of the microbial composition, defined as “dysbiosis”, is linked to a variety of autoimmune and inflammatory intestinal diseases including CeD [10,17]. 

Genetics, gestational age at birth, type of delivery, breastfeeding, lifestyle, diet, hormonal variations, and drug use (in particular the use of antibiotics and proton pump inhibitors) are the main factors regulating the composition of the gut microbiota [18,19].

As we have seen, CeD has a clear genetic component; however, the risk of developing celiac disease seems to be the result of a complex interplay between genetic predisposition and epigenetic factors, including a direct influence of gut microbial colonization. In fact, Sellitto et al. [20], monitoring breastfed children at risk of developing celiac disease (DQ2 haplotype carriers) for the first two years of life, found that these had a greater abundance of *Firmicutes* and *Proteobacteria* and a delay in the stabilization and maturation of the gut microbiota compared to controls not at risk of CeD, suggesting that the development of gut microbiota may be influenced by genetic factors related to the risk of CeD [20] This was also confirmed by a number of studies [21,22] through the prospective longitudinal study CDGEMM (Celiac Disease Genomic, Environmental, Microbiome, and Metabolomic Study), which enrolled 500 infants from the United States (USA), Italy, and Spain with the aim of identifying the risk factors associated with the onset of CeD. 

Wacklin et al. [23] showed that subjects with symptomatic CeD compared to asymptomatic ones had a different gut microbial composition, suggesting that the microbiota may also correlate with the clinical manifestations of CeD. In particular, the authors observed a higher abundance of *phylum* of *Proteobacteria* in symptomatic subjects compared to asymptomatic subjects who instead showed a more abundant increase in *phylum* of *Firmicutes*.

## 4. Early Environmental Factors, Microbiota and CeD

What early environmental factors can influence the development of CeD through gut microbial modulation? 

Premature birth leads to delayed colonial colonization, limited bacterial diversity, low bacterial load, reduction of obligate anaerobic commensals (*Bifidobacteria* and *Bacteroides*), and increase of facultative and pathogenic anaerobes (*Enterobacter*, *Enterococcus*, *Escherichia*, *Klebsiella*, *C. difficile* and *Staphylococus*), thus favoring an intestinal microbial structure of the inflammatory type that can facilitate CeD [24]. 

Birth modality is also considered a critical determinant in the early colonization of the neonatal gut microbiota. In fact, caesarean section involves a neonatal colonization completely of an environmental type and maternal skin derivation, with an increase in *Enterococcus faecalis* and a reduction in *Bacteroides* and *Parabacteroides* associated with an increased risk of celiac disease [25,26].

Tanpowpong et al. [27] studied 44,539 mother–infant pairs in order to test whether pregnancy and/or birth-related factors could be associated with CeD, documenting 173 infants (0.4%) diagnosed with CeD. The adjusted hazard ratio of cesarean section for CeD was 1.39 (95% CI: 0.99, 1.96, *p* = 0.06) compared to those born vaginally. 

Diet is also one of the major drivers that guide the composition of the gut microbiota and the mode of breastfeeding is fundamental with respect to its initial structuring; in fact, we know that direct breastfeeding is associated with maximum intestinal growth of *Bifidobacterium* spp. Cenit et al. [28] observed that the continuation of breastfeeding at the time of gluten introduction correlates with increased transmission to milk of IL-12p70, transforming growth factor-1 (TGF)-1, secretory IgA (sIgA), and *Bifidobacterium* spp., with reduced risk or delay in the development of CeD. It should be noted, however, that despite these observations and many previous ones in the same line, large epidemiological observation studies failed to document any protective effect of breast milk on the development of CeD in genetically predisposed children [29,30]. 

Numerous studies also associate early exposure to antibiotics (with a dose-dependent relationship) with the development of chronic autoimmune and inflammatory bowel diseases including CeD [31,32,33,34]. Especially relevant in this regard is the cohort study conducted by Dydensborg Sander S. et al. [32], which collected data from more than 1.7 million children born in Denmark and Norway, of which 3,346 were diagnosed with CeD, showing that exposure to antibiotics in the first year of life was positively associated with a later CeD diagnosis. 

Finally, infections can also act as triggering factors with respect to the risk of CeD, especially viral infections contracted in the first 2 years of life. *Rotavirus*, *Enterovirus, Adenovirus* type 12, and *Orthoreovirus* can trigger mimicry mechanisms with respect to gliadin, while *Orthoreoviruses*, through Toll-like 3 (TLR3) activation, can alter the stimulation of innate immunity and trigger intestinal inflammation and loss of tolerance to gliadin [1,35,36]. Recently, a cumulative effect augmenting the risk of developing CeD was demonstrated between *Enterovirus* and amount of gluten [37]. 

We will next focus on the causal role of specific bacterial groups in the pathogenesis of CeD. 

## 5. Gut Microbiota and the Pathogenesis of CeD

While gluten is the only necessary environmental factor, delaying its introduction into the diet at 12 months has been shown not to prevent the onset of CeD but only postponing its emergence [29]. However, the introduction of gluten modifies the gut microbiota of the host at risk of CeD. In fact, gluten-rich foods modify the microbiota especially in terms of the abundance of *phyla Firmicutes* and *Proteobacteria* [20]. In addition, recent studies show that, even though mammals lack proteases to digest gliadin, gluten metabolism is closely related to the microbiota of the entire oro-gastro-intestinal tract [8,10,11]. 

The gut microbiota can influence the digestion of gluten peptides by generating immunogenic peptides, but also, on the contrary, by cleaving immunogenic peptides undigested by human intestinal enzymes. These effects combine to impact the inflammatory immune response to gluten and the loss of gluten tolerance, ultimately therefore affecting the pathogenesis, progression, and clinical expression of CeD [18,28,38]. 

Compared to healthy individuals, patients with CeD have an altered composition (quantitative and qualitative) of the gut microbiota strictly dependent on genetics, age, and disease status. However, these changes could be both the cause and the consequence of the inflammatory state and the immune dysregulation that characterize CeD.

Several studies have shown a significant increase in CeD subjects compared to healthy ones of Gram-negative bacteria and in particular of *Bacteroides, E. coli, Enterobacteriaceae* and decrease of *Bifidobacterium*, *Streptococcus*, *Provetella* and *Lactobacillu* spp. [3,23,39,40,41]. 

Olivares et al. [42], following the children of the PROFICEL cohort for 5 years, confirmed a different composition of the fecal microbiota according to the type of breastfeeding (breastfeeding vs formula). In both cases, they detected in infants with the highest genetic risk for CeD an increase in enterotoxigenic species of *Escherichia coli* (ETEC), suggesting a possible association between genetic susceptibility to celiac disease and the presence of pathogenic bacteria in early life in the gut, predisposing to CeD development. 

Sanchez et al. [16] found higher diversity values for patients with active CeD compared to controls, as well as increased *phylum Proteobacteria* and decreased *phylum Firmicutes*. Members of the *Enterobacteriaceae* and *Staphylococcaceae*, particularly the species *Klebsiella oxytoca*, *Staphylococcus epidermidis,* and *Staphylococcus pasteuri*, were more abundant in patients with active disease than in controls. In contrast, members of the *Streptococcaceae* were less abundant in patients with active CeD than in controls. D’Argenio et al. [43] showed that members of the genus *Neisseria* (class *β-Proteobacteria*), later identified as *Neisseria flavescens* (CD-Nf), were significantly more abundant in patients with active CeD than in patients on GFD and controls. Also, an increase in the relative amounts of gram-negative bacterial genera such as *Bacteroides, Prevotella,* and *Escherichia* and reduced amounts of protective anti-inflammatory bacteria such as bifidobacteria and lactobacilli were confirmed in CeD patients [1,44]. In particular, these authors found significant reductions of *Bifidobacterium longum* in both feces and duodenal biopsies compared to controls, while the prevalence of *B. catenulatum* was higher in control biopsies than in both active and non-active celiac patients. 

What emerges from these data is that the gut microbiota, both in its commensal and pathogenic components, could influence the pathogenesis, progression, and clinical expression of CeD. In fact, the eubiotic microbiota, through the fermentation of dietary fiber, produces short-chain fatty acids (SCFAs), such as acetate, propionate, and butyrate, which are involved in a series of processes essential to the preservation of healthy gut immunity. Among them, the maturation of the intestinal mucosa, the modulation of mucin turnover, and the expression of tight junctions, thus ultimately modulating the intestinal permeability, the homeostasis of the intestinal mucosa, and the proper functions of the innate and adaptive immune system. Thus, it seems logical that a perturbation of the eubiosis is capable of creating an inflammatory state disrupting the integrity of the mucosal barrier and opening the way for gluten to cause—in predisposed individuals –the development of CeD. Recently, it has been found that the gut microbiota can contribute to the development of CeD also through the production of indole derivatives from tryptophan metabolism that act as agonists of the aryl hydrocarbon receptors (AhR) that are under-expressed in CeD subjects. Indole derivatives are in fact involved in the homeostasis of the immune system and in the inhibition of the activation of the actin regulatory protein MyoIIA that modulates the functional integrity of tight junctions and the permeability of the intestinal barrier [45,46]. In addition, the dysbiotic microbiota that characterizes CeD is rich, as mentioned, in Gram-negative microorganisms and this leads to an increase in lipopolysaccharides (LPS), released from the outer membrane of such bacteria; LPS increase the permeability of the intestinal barrier thus further allowing gliadin peptides to easily translocate from the intestinal lumen to the lamina propria, stimulating the release of IL-15, keratinocyte grow factor and IL-8, and activating both innate and adaptive immunity. 

## 6. The Role of Oral Microbiota

While, not surprisingly, most studies investigating the role of microbiota in CeD have focused on the gut microbiota, the salivary microbiota (consisting of more than 700 bacterial species) has begun to attract the interest of researchers with respect to the pathogenesis not only of CeD, but also of other pathologies [47,48,49].

There are about 10^11^ bacteria per gram of dental plaque and 10^8^ bacteria per milliliter of saliva. Adults produce more than 1000 mL of saliva every day which is continuously swallowed in the gastrointestinal tract. Therefore, oral microorganisms can become an important reservoir of bacteria in the gut and can play an important role in maintaining the internal stability of the gut microecosystem. 

The latest research shows that oral symbiotic microorganisms participate in the immune function of the oral mucosa and in preventing the invasion of pathogens. For example, the genera *Veillonella* and *Streptococcus* promote the production of antimicrobial peptides and the secretion of inflammatory cytokines, thus increasing the epithelial barrier function and the oral mucosal thickness. However, other studies show that some oral pathogenic bacteria are not only related to oral diseases such as dental caries, periodontitis, and oral ulcers but, through intestinal ectopic colonization, they are also related to cardiovascular disease, obesity, diabetes, rheumatoid arthritis, intestinal diseases, cancer, Alzheimer’s disease, preterm birth, etc. [8,10,50].

Fernandez-Feo et al. [11] isolated microorganisms from oral plaque and saliva that can be grown and are able to digest gluten: *Rothia mucilaginosa* HOT-681, *Rothia aeria* HOT-188, *Actinomyces odontolyticus* HOT-701, *Streptococcus mitis* HOT-677, *Streptococcus* sp. HOT-071, *Neisseria mucosa* HOT-682, and *Capnocytophaga sputigena* HOT-775 and some *Lactobacilli* spp., in particular *Lactobacillus rhamnosus.* Whether this might bear future implications in helping protect the small intestinal mucosa from the harmful effects of ingested gluten remain clearly only speculative at this stage. 

## 7. Gluten-Free Diet and Gut Microbiota

As is well known, the gluten-free diet (GFD) is currently the only effective and indispensable treatment for CeD. While it is typically very effective, especially in pediatric age, up to 25–50% of patients fail to show a significant clinical improvement. Hence, current research is actively working to find alternative and additional treatments for CeD [51]. 

Several authors have studied GFD-induced changes in gut microbiota. Bonder et al. [52] demonstrated consistent changes induced by GFD in terms of gut microbial composition mainly in the *Veillonellaceae* family that tends to decrease together with *Ruminicoccus bromii*, and *Roseburia*, *while* families of *Victivallaceae, Clostridiaceae*, and *Coriobacteriaceae* increased.

Rinninella et al. [24], studying the impact of individual food components (macronutrients and micronutrients), salt, and food additives in different eating styles, concluded that GFD leads to a net reduction in microbial α-diversity and a decrease in beneficial microorganisms such as *Bifidobacterium* and *Lactobacillus*. Consequences of these changes are the collapse of intestinal production of SCFAs, modification of intestinal pH, enhanced intestinal permeability, and compromise of the host’s metabolic and immune functions. They also noted a parallel increase in abundance of harmful species such as *Enterococcus, Staphylococcus, Salmonella, Shigella*, and *Klebsiella*. 

Of interest, it appears that GFD not only fails to restore eubiosis of the digestive tract in CeD patients, but also negatively affects microbial homeostasis in healthy individuals [8].

Unfortunately, only scarce data are available on colonic or fecal microbiota in CeD patients *before* initiating the GFD and then prospectively followed in order to assess changes intervened *after* initiating the diet. 

Furthermore, it should be noted that the *Western* GFD diet is based on the prevalent use of ultra-processed and refined foods with a high fat and sugar content, as well as a low intake of dietary fiber, folic acid, iron, calcium, selenium, magnesium, zinc, niacin, biotin, riboflavin, pyridoxine, and vitamin D. Western GFD in a certain sense emphasizes the shortcomings of the western diet, and this appears to be quite detrimental for CeD patients who should instead lean toward a Mediterranean diet by increasing the use of organic, fresh foods, respecting seasonality and foods rich in fiber, microelements and bioactive vitamins. In this regard, it would be appropriate to increase the use of pseudocereals such as quinoa, amaranth, and sorghum and gluten-free cereals that are rich in fiber, minerals, thiamine, carotenoids, flavones, tannins, proteins, and healthy fats [9]. Also, nutraceutical supplementation, including the targeted use of probiotics, might be a winning proposition [53]. 

## 8. Targeted Microbiota Therapy for CeD

Pecora et al. [54] reviewed all the studies reported in Pubmed on the topic from November 2009 to November 2019. Their analysis showed that the gut microbiota may indeed play a decisive role with respect to gluten metabolism, modulation of the immune response, and modulation of the permeability of the intestinal barrier. The same has also been confirmed more recently by Rossi et al. [1] and Yemula et al. [19] arguing that probiotics, prebiotics, postbiotics, and fecal microbiota transplantation (FMT), together with the GFD, could contribute to functionally modifying the gut microbiota. The targeted microbiota intervention is essentially aimed at restoring the beneficial and SCFAs-producing obligate anaerobic commensal species, capable of reducing the permeability of the intestinal barrier and promoting immune homeostasis and counteracting pathogenic and inflammatory species. We will now briefly review the evidence on possible therapeutic options presented by these potential interventions. Table 1 shows clinical trials and in vitro studies testing probiotics in the treatment of CeD. (see Table 1)

## 9. Lactobacilli

Francavilla et al. [55] tested 18 probiotic strains of lactobacilli for peptidase activity (aminopeptidase N, imminopeptidase, prolyl endopeptidyl peptidase, tripeptidase, prolidase, prolinase, and dipeptidase), and verified that the simultaneous use of 10 strains (*Lactobacillus casei* BGP93, *Lactobacillus delbrueckii* subsp. *bulgaricus* SP5, *Lactobacillus paracasei* LPC01, and BGP2, *Lactobacillus plantarum* BGP12, LP27, LP35, LP40, LP47, and SP1) provided the peptidase pool necessary to completely degrade immunogenic gluten peptides including 33-mer peptide, peptide residues 57 to 68 of a9-gliadin, peptide A-gliadin 62–75, and peptides 62–75 of g-gliadin. The efficacy of lactobacilli strains in gluten digestion has also been demonstrated in vivo by challenging CeD patients in remission with administration for 60 days of bakery products containing gluten predigested by probiotics. No worsening of serological parameters or intestinal permeability were observed during the duration of the challenge, suggesting that probiotic endopeptidases were indeed able to completely degrade gluten and nullify its toxicity.

Jenickova et al. [56] examined in a randomized, double-blind study the impact of *Lactiplantibacillus plantarum* HEAL9 and *Lacticaseibacillus paracasei* 8700:2 on the fecal metabolome of children with CeD and found a significant change in the fecal metabolome in the intervention group compared to the control group, especially in the acid and amino acid profiles. In particular, they observed an increase in 4-hydroxyphenylacetate, which is a catabolite of the microbial proteolysis of tyrosine with a hepatoprotective effect and antioxidant properties, and a decrease in the concentrations of various amino acids and especially of threonine which is a very abundant amino acid in the protein nucleus of mucin, constitutive element of the intestinal barrier.

Hou et al. [65] showed in intestinal organoids that the *L. reuteri* D8 strain produced AhR ligands correlated with increased IL-22 production, intestinal stem cell proliferation, and restoration of intestinal mucosal damage. Since the microbiota (as we described earlier) also participates in the production of AhR agonists involved in the regulation of the immune system and in maintaining the functional integrity of the intestinal barrier, it is reasonable that the microbiota could be modulated also through a diet enriched with tryptophan or through the use of precision probiotics such as *Limosilactobacillus reuteri*, which is able to produce AhR ligands [45,46]. 

## 10. Bifidobacteria

A number of studies have been performed in in vitro system as well as both in animals and in humans with various strains of Bifidobacteria. Olivares et al. [57], from the same group that had previously demonstrated an anti-inflammatory effects of *Bifidobacterium longum* CECT 7347 in a mouse model of gliadin-induced enteropathy [58], investigated in a double-blind, randomized placebo controlled trial the potential effects of 3-month administration of the same probiotic in 33 children newly diagnosed with CeD. The authors reported in the probiotic group an improvement in health status, an increase in the speed of height growth and a reduction in several indexes of intestinal inflammation: decreased peripheral CD3(+) T lymphocytes and slightly reduced TNF-α.

Quagliariello et al. [59] administered two strains of *Bifidobacterium breve* (B632 and BR03) to CeD children in GFD for 3 months and found an increase in *Actinobacteria* and the restoration of the physiological *Firmicutes/Bacteroidetes* ratio compared to the placebo group.

McCarville et al. [61] analyzed the role of the *B. longum* NCC2705 strain, isolated from a healthy infant, in a gluten-sensitive NOD/DQ8 mouse model, and demonstrated the beneficial effect of the probiotic mediated by the production of a serine protease inhibitor (serpin) that exhibits immunomodulatory properties capable of reducing inflammation induced by gliadin exposure.

Primec et al. [60] carried out a double-blind controlled study aimed at identifying correlations between fecal microbiota, TNF-α and SCFAs in healthy children and children with CeD after administration of the probiotic *Bifidobacterium breve* BR03 and B632. The effect of probiotic administration showed a negative correlation between *Verrucomicrobia, Synergistetes,* and *Euryarchaeota* which could play a role in the intestinal anti-inflammatory process.

## 11. Combining Lactobacilli and Bifidobacteria

Francavilla et al. [62] completed a prospective randomized study on 109 CeD patients on GFD who nevertheless experienced symptoms compatible with irritable bowel syndrome (IBS). The subjects received a multistrain probiotic mixture containing *Lactobacillus casei* LMG 101/37 P-17504, *Lactobacillus plantarum* CECT 4528, *Bifidobacterium animalis* subsp. *lactis* Bi1 LMG P-17502, *Bifidobacterium breve* Bbr8 LMG P-17501, and *Bl. breve* Bl10 LMG P-17500 or placebo for six weeks. The patients on the probiotic mixture reported a reduction in IBS symptoms associated with an increase in lactic acid-producing bacteria such as staphylococci and bifidobacteria. 

Tremblay et al. [63] tested the role of a multi-strain probiotic mixture containing *Lactobacillus helveticus* Rosell-52, *Bifidobacterium infantis* Rosell-33 and *Bifidobacterium bifidum* Rosell-71 with fructooligosaccharides in children with CeD on GFD who had persistence of gastrointestinal symptomatology. The children on the probiotic mixture showed significantly reduced symptomatology compared to the placebo group. 

Lastly, Lionetti et al. [64] verified the efficacy of a multispecies probiotic (*Lactobacillus paracasei* 101/37 LMG P-17504, *Lactobacillus plantarum* 14D CECT 4528, *Bifidobacterium animalis* subsp. *lactis* Bi1 LMG P-17502, *Bifidobacterium breve* Bbr8 LMG P-17501, *Bifidobacterium breve* BL10 LMG P-17500) administered for 12 weeks in 96 newly diagnosed CeD children on GFD. Children in both, the probiotic group and in the placebo group showed a significant increase in body max index (BMI) score after 3 and 6 months of treatment, but the increase in BMI-Z score was significantly higher and faster in the probiotic group. However, no other differences were noted in all other clinical and laboratory parameters.

## 12. Postbiotics

Postbiotics, also called “ghost probiotics”, are defined as the preparation of inanimate microorganisms and/or their components or bioactive products that confer health benefit to the host. 

Freire et al. [67] tested the role of postbiotics (butyrate, lactate, and polysaccharide A produced by *B. fragilis*) on models of intestinal organoids developed from duodenal biopsies of celiac and non-celiac patients showing that they can improve intestinal barrier function through increased TJ expression. In addition, these microbiota-derived molecules were able to reduce gliadin-induced cytokine secretion.

Conte et al. [68] investigated the in vitro effect on Caco-2 cells of the *Lactobacillus paracasei* CBA L74 postbiotic with respect to the prevention of gliadin-induced activation of inflammatory responses. The postbiotic was able to induce autophagy in Caco-2 cells and prevent the inflammatory effects of gliadin.

## 13. Conclusions and Future Directions

As we have seen, the main conundrum in all of this is the fact that we still do not fully understand whether the changes observed in the microbiota of CeD subjects are the consequences or part of the causes of the disease. To unravel this mystery, it would be appropriate to construct longitudinal studies that prospectively follow subjects at risk of CeD for a long time, possibly from birth until the development of CeD autoimmunity and then to the full-blown disease, and then again after the institution of a proper GFD so that one can mechanistically link the variations in the intestinal microbial composition with the pathogenesis and clinical expression of the disease. These modifications take place specifically in the phase of passage from tolerance to the loss of tolerance to gluten. The availability of such data from a large population in order to account for individual variabilities would allow to identify the specific causal links between microbial composition and function on one side and disease on the other, thus leading to the identification of new targets for precise epigenetic modulation aiming at the prevention and treatment of CeD. 

So far, few cross-sectional and longitudinal studies have been formulated in this way. Sellitto et al. [20], as well as Olivares et al. [69], showed that the composition of the microbiota of infants at genetic risk of developing CeD is different from that of non-at-risk infants. In fact, in children at risk they observed a delay in the maturation trajectory of the microbiota, a reduced amount of *Bacteroidetes* and an excess of *Proteobacteria* and *Firmicutes* (in particular of the genre *Lactobacillus*) already evident before the appearance of CeD autoimmunity. 

Subsequently, in a series of papers, Leonard et al. [4,21,22,70] published data from a prospective observational study using the Celiac Disease, Genomic Environmental Microbiome, and Metabolomic study (CeDGEMM) cohort aimed at identifying the existence of a specific microbial and metabolomic signature capable of predicting loss of gluten tolerance in relation to environmental risk factors and genetic markers of susceptibility to CeD. The authors demonstrated, in subjects who subsequently developed the disease, a decrease in the abundance of anti-inflammatory microbial species (*Streptococcus thermophilus*, *Faecalibacterium prausnitzii* and *Clostridium clostridioforme*) and an increase in inflammatory species (*Dialister invisus*, *Parabacteroides* sp., *Lachnospiraceae*) related and increased metabolism of tryptophan and the metabolites serine and threonine. Of note, such dysbiosis was not detectable in children who, despite showing the same risk factors, did not subsequently develop the disease. 

Very recently, the same “CDGEMM Team” [66] identified, isolated, cultured, and sequenced a novel strain of *B. vulgatus* (20220303-A2) present only in subjects with risk factors for CeD but who did not develop the disease. This strain, when its cell-free supernatant was tested in a human gut organoid system developed from pre-celiac patients, was able to mitigate the effects of gliadin exposure on intestinal epithelial homeostasis by epigenetically reprogramming the mechanisms controlling intestinal permeability, immune response, and cell repair phenomena. These findings raise the possibility that this unique strain of *B. vulgatus* may play a protective role with respect to the risk of loss of gluten tolerance in subjects at risk for CeD.

## Figures and Tables

**Figure 1 nutrients-16-01882-f001:**
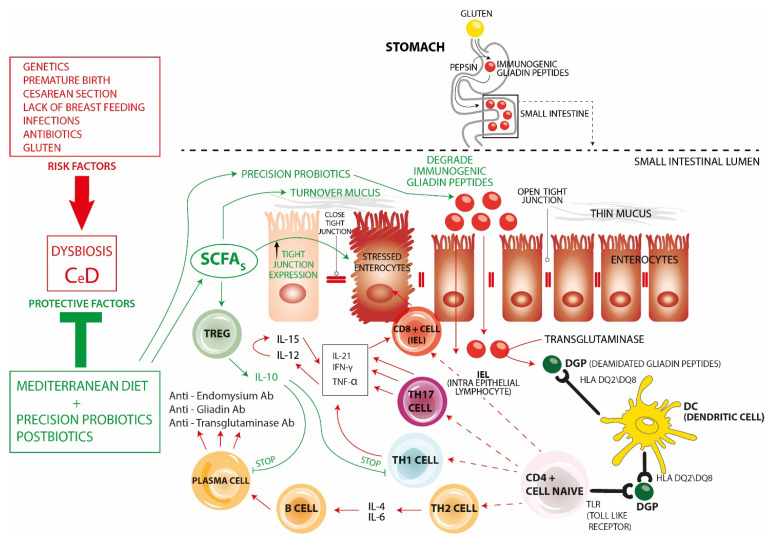
Pathophysiology of CeD and Targeted Microbiota Intervention.

**Table 1 nutrients-16-01882-t001:** Probiotics tested to treat Celiac Disease.

(A) Clinical Trials	
Probiotics Tested to Treat Celiac Disease	Clinical Trials
*Lactobacillus casei* BGP93, *Lactobacillus delbrueckii* subsp. *bulgaricus* SP5, *Lactobacillus paracasei* LPC01 and BGP2, *Lactobacillus plantarum* BGP12, LP27, LP35, LP40, LP47 and SP1	[55]
*Lactiplantibacillus plantarum* HEAL9 and *Lacticaseibacillus paracasei* 8700:2	[56]
*Bifidobacterium longum* CECT 7347 (ES1)	[57]
	[58]
*Bifidobacterium breve* (B632 and BR03)	[59]
	[60]
*B. longum* NCC2705	[61]
*Lactobacillus casei* LMG 101/37 P-17504, *Lactobacillus plantarum* CECT 4528, *Bifidobacterium animalis* subsp. *lactis* Bi1 LMG P-17502, *Bifidobacterium breve* Bbr8 LMG P-17501 and *Bl. breve* Bl10 LMG P-17500	[62]
*Lactobacillus helveticus* Rosell-52, *Bifidobacterium infantis* Rosell-33 and *Bifidobacterium bifidum* Rosell-71 with fructooligosaccharides	[63]
(*Lactobacillus paracasei* 101/37 LMG P-17504, *Lactobacillus plantarum* 14D CECT 4528, *Bifidobacterium animalis* subsp. *lactis* Bi1 LMG P-17502, *Bifidobacterium breve* Bbr8 LMG P-17501, *Bifidobacterium breve* BL10 LMG P-17500)	[64]
**(B) In vitro studies**	
**Probiotics tested to treat Celiac Disease**	**In vitro studies**
*L. reuteri* D8	[65]
*B. vulgatus* (20220303-A2)	[66]

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
