# Peer review of "How the Microbiota May Affect Celiac Disease and What We Can Do"

_nutrients, 2024, doi:10.3390/nu16121882_

Round 1

Reviewer 1 Report

Comments and Suggestions for Authors

In this review the authors discussed the effect of microbiota and nutrition on the pathogenesis of celiac disease. The review included the following points: a) The gut microbiota and Celiac disease. how the microbiota differs among populations based on genetics, enviornmental and other factors.  b) The role of oral microbiota and gut microbiota in the pathogenesis of celiac diseases.  c) The effect of glutien free diet on the structure and distribution of microbiota.   d) Use of microbiotic as drugs such as lactobacillus and biofidobacterium

Although the review is comperhensive, the idea is not noval, and previously published 

PMID: 32499787

PMID: 26725064

PMID: 36980164

PMID: 32651763

Major concerns

1) What are new directions or findings in this review that are not published previosuly?

2) I wound suggest the authors to include tables summerizing the trials used by microbiota to treat celiac diseases showing both in vitro and in vivo model.

3) The mechanisms of microbiota inducing celiac disease should be summerized in tables or figures showing all signaling pathways associated with microbiota-mediated disease pathogenesis 

Comments on the Quality of English Language

Moderate editing

Author Response

Thank you for your comments.

  1. While it is indeed true that other reviews have been published, it is also true that this is a fluid and highly moving target, and substantial new information has been gathered since most of the past reviews have been published. In fact, only 1 of the reviews indicated by the reviewer has been published after 2020 (PMID: 36980164) and even this recent review (that we properly inserted in the bibliography) cannot mention what appear a major breakthrough study (PMID: 38177249). In essence, we still believe that our review is a needed update to the field.
  2. As requested by the reviewer, a Table has been added including clinical trials and in vitro studies with probiotics tested to treat celiac disease.
  3. We actually think that the mechanisms of microbiota inducing celiac disease have been summarized in our figure.
  4. Respectfully, we do not feel that our paper is in need of "moderate English revision"

We again thank this reviewer and believe we have satisfactorily addressed all of his/her remarks.

Reviewer 2 Report

Comments and Suggestions for Authors

nutrients-3045257_ How the microbiota may affect celiac disease and what we can do.

This review article is submitted to the "Prebiotics and Probiotics" section of the journal.

It is a well-written review article.

Comments:

  • The title of the work is appropriate to its content.
  • The abstract clearly and precisely outlines the hypothesis and objective. However, considering that the abstract is the part of the article that remains accessible on platforms for potential readers to select, I believe it would be improved by specifying the type of review conducted, the time period covered by the review, which allows for connections with past and future reviews in the scientific knowledge of this topic. Additionally, it should mention the databases used in the search for articles and the number of articles included in the review. Finally, a sentence indicating the main conclusion of the work should be added. This would present the content of the work more precisely.
  • The introduction is clear and precise, using relevant bibliography. However, in the last section, which states the objective of this review, the period covered by the review should be specified, and bibliography should not be used in the objective.
  • The results are structured into sections that facilitate a better understanding of the topic.
  • The figure synthesizes the knowledge on the topic.
  • Given the importance of the topic, I suggest including a material and methods and conclusions sections.

Author Response

We thank Reviewer 1 for their thorough review of our paper, their positive comments and important recommendations that we tried to address to the best of our capacity.

  • In the abstract: As requested, we have indicated the time period covered buy the review, the database searched, the number of articles identified and added a final sentence indicating the main conclusions of the review
  • In the Introduction, we have added the timeframe of our review. Also, as requested we eliminated the bibliography from the objectives.
  • Lastly, the final section has been more appropriately re-defined as "conclusions and future directions"

Round 2

Reviewer 1 Report

Comments and Suggestions for Authors

No further comments

Comments on the Quality of English Language

Fine